# Black Box Probabilistic Numerics

**Onur Teymur**
University of Kent
Alan Turing Institute

**Christopher N. Foley**
University of Cambridge
Optima Partners

**Philip G. Breen**
Roar AI

**Toni Karvonen**
University of Helsinki
Alan Turing Institute

**Chris. J. Oates**
Newcastle University
Alan Turing Institute

## Abstract

*Probabilistic numerics* casts numerical tasks, such the numerical solution of differential equations, as inference problems to be solved. One approach is to model the unknown quantity of interest as a random variable, and to constrain this variable using data generated during the course of a traditional numerical method. However, data may be nonlinearly related to the quantity of interest, rendering the proper conditioning of random variables difficult and limiting the range of numerical tasks that can be addressed. Instead, this paper proposes to construct probabilistic numerical methods based only on the final output from a traditional method. A convergent sequence of approximations to the quantity of interest constitute a dataset, from which the limiting quantity of interest can be extrapolated, in a probabilistic analogue of Richardson's deferred approach to the limit. This *black box* approach (1) massively expands the range of tasks to which probabilistic numerics can be applied, (2) inherits the features and performance of state-of-the-art numerical methods, and (3) enables provably higher orders of convergence to be achieved. Applications are presented for nonlinear ordinary and partial differential equations, as well as for eigenvalue problems—a setting for which no probabilistic numerical methods have yet been developed.

## 1 Introduction

*Probabilistic numerics* (PN) has attracted significant recent interest from researchers in machine learning, motivated by the possibility of incorporating probabilistic descriptions of *numerical uncertainty* into applications of probabilistic inference and decision support [1, 2]. PN treats the intermediate calculations performed in running a traditional (*i.e.* non-probabilistic) numerical procedure as *data*, which can be used to constrain a random variable model for the quantity of interest [3]. Conjugate Gaussian inference has been widely exploited, with an arsenal of PN methods developed for linear algebra [4–11], cubature [12–30], optimisation [31–36], and differential equations [37–55]. However, nonlinear tasks pose a major technical challenge to this approach, as well as to computational statistics in general, due to the absence of explicit conditioning formulae. Compared to traditional numerical methods, which have benefited from a century or more of sustained research effort, the current scope of PN is limited. The performance gap is broadly characterised by the absence of certain important functionalities—adaptivity, numerical well-conditioning, efficient use of computational resource—all of which contribute to limited applicability in real-world settings.

This article proposes a pragmatic solution that enables state-of-the-art numerical algorithms to be immediately exploited in the context of PN. The idea, which we term *black box probabilistic numerics* (BBPN), is a statistical perspective on *Richardson's deferred approach to the limit* (RDAL) [56]. The starting point for BBPN is a sequence of increasingly accurate approximations produced by a

traditional numerical method as its computational budget is increased. Extrapolation of this sequence (to the unattainable limit of 'infinite computational budget') is formulated as a prediction task, to which statistical techniques can be applied. For concreteness, we perform this prediction using *Gaussian process*es (GPs) [57], but other models could be used. Note that we do not aim to remove the numerical analyst from the loop; the performance of BBPN is limited by that of the numerical method on which it is based.

There are three main advantages of BBPN compared to existing methods in PN: (1) BBPN is applicable to any numerical task for which there exists a traditional numerical method; (2) state-of-the-art performance and functionality are automatically inherited from the underlying numerical method; (3) BBPN achieves a provably higher order of convergence relative to a single application of the numerical method on which it is based, in an analogous manner to RDAL. The main limitations of BBPN, compared to existing methods in PN, are: (1) multiple realisations of a traditional numerical method are required (*i.e.* one datum is not sufficient in an extrapolation task), and (2) a joint statistical model has to be built for not just the quantity of interest (as in standard PN), but also for the error associated with the output of a traditional numerical method. The capacity of generic statistical models, such as GPs, to learn salient aspects of this structure from data and to produce meaningful predictions over a range of real-world numerical problems, demands to be investigated.

The article is organised as follows: In Section 2 we recall classical RDAL. In Section 3 we lift RDAL to the space of probability distributions, exploiting GPs to instantiate BBPN and providing a theoretical guarantee that higher order convergence is achieved by using GPs within BBPN. In Section 4 we present a detailed empirical investigation into BBPN, demonstrating its effectiveness on challenging tasks that go beyond the capability of current methods in PN, while also highlighting potential pitfalls. As part of this we perform a comparison of the uncertainty quantification properties of BBPN against earlier approaches. A closing discussion is contained in Section 5.

## 2   Turning Lead into Gold

Our starting point is the celebrated observation of Richardson [56], that multiple numerical approximations can be combined to produce an approximation more accurate than any of the individual approximations. To see this, consider an intractable scalar quantity of interest $q^* \in \mathbb{R}$, and suppose that $q^*$ can be approximated by a numerical method $q$ that depends on a parameter $h > 0$, such that

$$q(h) = q^* + Ch^\alpha + \mathcal{O}(h^{\alpha+1}) \tag{1}$$

for some $C \in \mathbb{R}$ (which may be unknown) and $\alpha > 0$ (which is assumed known, and called the *order* of the method). Clearly $q(h)$ converges to $q^*$ as $h \to 0$, but we also suppose that the *cost* of computing $q(h)$ increases in the same limit, with exact evaluation of $q(0)$ requiring a hypothetically infinite computational budget. Proposition 1 is the cornerstone of RDAL. It demonstrates that two evaluations of a numerical method of order $\alpha$ can be combined to obtain a numerical method of order $\alpha + 1$. An elementary proof of this foundational result is provided in Appendix A.

**Proposition 1.** *Let $q$ be a numerical method of order $\alpha$, as in (1). Fix $\gamma \in (0,1)$ and let $q_\gamma(h)$ denote the height at which a straight line drawn through the points $(h^\alpha, q(h))$ and $((\gamma h)^\alpha, q(\gamma h))$ intersects the vertical axis in $\mathbb{R}^2$. Then $q_\gamma$ is a numerical method of order $\alpha + 1$.*

Now consider the natural generalisation of Proposition 1, in which we compute approximations $q(h_i)$ along a decreasing sequence of values $(h_i)_{i=1}^n$. One can then fit a smooth interpolant to the points $\{(h_i^\alpha, q(h_i))\}_{i=1}^n$ (generalising the straight line through two points), then extrapolate this to $h = 0$, to give an estimate for the quantity of interest. This simple idea is widely used in numerical analysis; its potential to radically improve solution accuracy, given only a sequence of simple calculations as input, prompted Press et al. [58, p. 922] to describe it as "turning lead into gold". The practical success of RDAL depends on the choice of interpolant, with polynomial interpolation being most commonly used. Unqualified, RDAL is usually understood to refer to an order $n-1$ polynomial fitted to $n$ points, which produces a numerical method of order $\alpha+n$; see Theorem 9.1 of [59]. Higher-order polynomial extrapolation is known to perform poorly unless the values $(h_i)_{i=1}^n$ are able to be chosen specifically to mitigate Runge's phenomenon [60], motivating the Bulirsch–Stoer algorithm [61], which instead fits a rational function interpolant. This allows both greater expressiveness and robustness than polynomial interpolation (though not necessarily as efficiently [58]). These methods are all situated within the broad category of *extrapolation methods* in numerical analysis; a comprehensive historical survey can be found in [62].

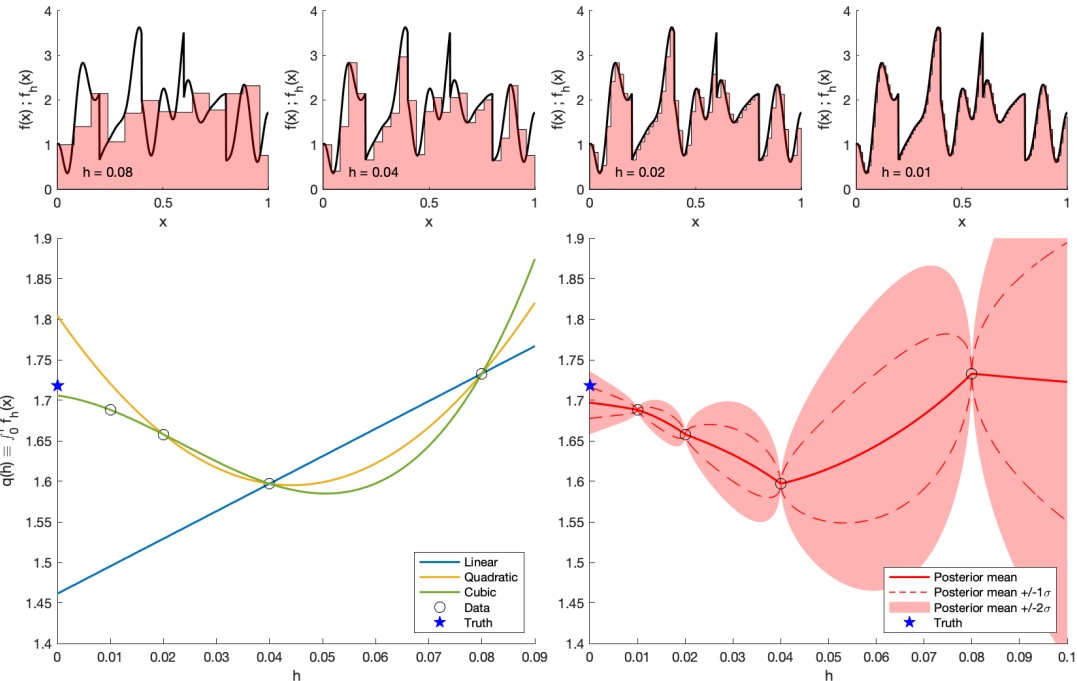

Figure 1: Richardson's deferred approach to the limit (RDAL), applied to the method of Riemann sums with integration bandwidth $h$ and an oscillatory integrand $f(x)$, displayed in the four top panes. The bottom left pane shows linear, quadratic and cubic interpolants converging on the true value of the integral, denoted by a blue star. The strength of RDAL is seen in the fact that the cubic interpolant gives a better estimate than that given by the finest-grid Riemann sum with $h = 0.01$. (This example is closely related to Romberg's method.) The bottom right pane illustrates *black box probabilistic numerics* (BBPN), in which a Gaussian process is fitted to the same data. The GP specification is crucial to the performance of BBPN, and is described in Section 3.

Figure 1 presents a simple visual demonstration of RDAL, applied to the method of Riemann sums for an oscillatory 1D integrand. While RDAL gives improved approximations, no quantification of estimation uncertainty is provided. The only attempt of which we are aware to provide uncertainty quantification for RDAL is due to [63], who focused on the Navier–Stokes equation and a specific scalar quantity of interest. Here we go further, proposing the general framework of BBPN and introducing novel methodology that goes beyond scalar quantities of interest. The right-hand pane of Figure 1 displays the outcome of the method we are about to introduce, applied to the same task—observe that the true value of the integral falls within the $\pm 2\sigma$ credible set produced using BBPN. Details of the simulations in this figure are contained in Appendix C.1.

## 3   Methodology

The core idea of BBPN is to model $q(h)$ as a *stochastic process* $Q(h)$ rather than fit a deterministic interpolant as in RDAL. The distribution of the marginal random variable $Q(0)$ is then interpreted as a representation of the epistemic uncertainty in the quantity of interest $q(0)$. Conjugate Gaussian inference can be performed in BBPN, since one needs only to construct an interpolant. This means the challenge of nonlinear conditioning encountered in PN [4] is avoided, massively extending the applicability of PN. In addition to being able to leverage state-of-the-art numerical methods, the BBPN approach enjoys provably higher orders of convergence relative to a single application of the numerical method on which it is based; see Section 3.3.

### 3.1   Notation and Setup

Our starting point is to generalise (1) to encompass essentially all numerical tasks, following the abstract perspective of [64]. To do so, we observe that any quantity of interest $q^*$ can be characterised by a sufficiently large collection of real values $q^*(t)$, with $t$ ranging over an appropriate index set $T$.

**Definition 2.** *A traditional numerical method is defined as a map $q : [0, h_0) \times T \to \mathbb{R}$, for some $h_0 > 0$ such that, for all $t \in T$, the function $h \mapsto q(h, t)$ is continuous at 0 with limit $q(0, t) = q^*(t)$.*

For example, a (univariate) *initial value problem* (IVP), in which $t$ is interpreted as time, can be solved using a traditional numerical method $q$ whose time step size $h$ trades off approximation error against computational cost. The output $q(h, t)$ of such a method represents an approximation to the true solution $q^*(t)$ of the IVP, at each time $t$ for which the solution is defined. In general, depending on the numerical task, the index $t$ could be spatio-temporal, discrete, or even an unstructured set, while the meaning of the index $h$ will depend on the numerical method.

Definition 2 thus encompasses, among other things: (1) adaptive integrators for time-evolving *partial differential equation*s (PDEs), where $h > 0$ represents a user-specified error tolerance, and the spatio-temporal domain of the solution is indexed by $T$; (2) iterative methods for approximating the singular values of a $d \times d$ matrix, where for example $h := w^{-1}$ with $w$ the number of iterations performed, and the ordered singular values are indexed by $T = \{1, \ldots, d\}$; and (3) the simultaneous approximation of multiple related quantities of interest, where $T$ indexes not only the domain(s) on which the individual quantities of interest are defined, but also the multiple quantities of interest themselves (this situation arises, for example, in inverse problems that are *PDE-constrained* [65]).

The perspective in Definition 2 is abstract but, as these examples make clear, it will typically only be possible to compute $q$ at certain input values (such as $h = w^{-1}$ for $w \in \mathbb{N}$, or for just a finite collection of inputs $t$ if the index set $T$ is infinite), and furthermore each evaluation is likely to be associated with a computational cost. Thus complete information regarding the map $q : [0, h_0) \times T \to \mathbb{R}$ will not be available in general, and there will therefore remain *epistemic uncertainty* in its complete description. Our aim in Section 3.2, in line with the central philosophical perspective of PN, is to characterise this uncertainty using a statistical model.

## 3.2 Black Box Probabilistic Numerics

The proposed BBPN approach begins with a *prior* stochastic process $Q$ and constrains this prior using data $D$. Concretely, we assume that the real values $q(h_i, t_{i,j})$ are provided at a finite set of resolutions $h_1 > \cdots > h_n > 0$ and distinct ordinates $t_{i,1}, \ldots, t_{i,m_i} \in T$. Note that the number of $t$-ordinates $m_i$ can depend on $h_i$. Our dataset therefore contains the following information on $q$:

$$D := \{(h_i, t_{i,j}, q(h_i, t_{i,j})) : i = 1, \ldots, n; \ j = 1, \ldots, m_i\} \tag{2}$$

The stochastic process obtained by conditioning $Q$ on the dataset $D$, denoted $Q|D$, implies a marginal distribution for $Q(0, \cdot)$, which we interpret as a statistical prediction for the unknown quantity of interest $q^*(\cdot)$. In order for uncertainty quantification in this model to be meaningful, one either requires expert knowledge about the numerical method that generated $D$, or one must employ a stochastic process that is able to adapt to the data, so that its predictions can be calibrated.

Our goal is to specify a stochastic process model $Q(h, t)$ that behaves in a desirable way under extrapolation to $h = 0$. To this end, we decompose

$$Q(h, t) = Q^*(t) + E(h, t) \tag{3}$$

where $Q^*(t)$ is a prior model for the unknown quantity of interest $q^*(t)$, and $E(h, t)$ is a prior model for the error of the numerical method. It will be assumed that $Q^*$ and $E$ are independent (denoted $Q^* \perp\!\!\!\perp E$), meaning that prior belief about the quantity of interest is independent of prior belief regarding the performance of the numerical method. (This assumption is made only to simplify the model specification, but if detailed insight into the error structure of a numerical method is available then this can be exploited.) Compared to the existing PN methods cited in Section 1, a prior model for the error $E$ is an additional requirement in BBPN.

The error $E(h, t)$ is assumed to vanish[1] as $h \to 0$, meaning that a stationary stochastic process model for $E(h, t)$, and hence for $Q(h, t)$, is inappropriate, and can result in predictions that are both severely biased as well as under-confident; see Appendix C.2. In the next section, we propose a parsimonious non-stationary GP model for $Q(h, t)$ of the form (3), which combines knowledge of

---

[1]This statement covers several potentially subtle notions from numerical analysis such as well-posedness of the problem and numerical stability of the algorithm; these are studied in detail in their own right in the literature, and for our purposes it suffices to assume that the error behaves well in the limit.

the *order* of the numerical method (only) with data-driven estimation of GP hyperparameters. This setting is practically relevant—the order of a numerical method is typically one of the first theoretical properties that researchers aim to establish while, conversely, for more complex numerical methods the order may actually be the only salient high-level error-characterising property that is known, and thus represent the limit of mathematical insight into the method.

### 3.3 Gaussian Process BBPN

Gaussian processes provide a convenient model for $Q^*$ and $E$, since they produce an explicit form for the conditional $Q|D$. The details of conjugate Gaussian inference are standard (see e.g. [57]) and so relegated to Appendix B.1; our focus here is on the specification of GP priors for $Q^*$ and $E$.

The notation $Q \sim \mathcal{GP}(\mu_Q, k_Q)$ will be used to denote that $Q$ is a GP with mean function $\mu_Q$ and covariance function $k_Q$. With no loss of generality, in what follows we consider centred processes (*i.e.* $\mu_Q = 0$). It will be assumed that $T = T_1 \times \cdots \times T_p$, with each $T_i$ either a discrete or a continuous subset of a Euclidean space, with the Euclidean distance between elements $t_i, t_i' \in T_i$ being denoted $\|t_i - t_i'\|$. (Typical applications involve small $p$; for example, the domain of a spatio-temporal PDE is typically decomposed as $T = T_1 \times T_2$ where $T_1$ indexes time and $T_2$ indexes all spatial dimensions, so that $p = 2$.)

**Prior for $Q^*$:**   In the absence of detailed prior belief about $q^*$, we consider the following default prior model. Let $G \sim \mathcal{GP}(0, \sigma^2 \rho_G k_G)$, $Z = (Z_1, \ldots, Z_v) \sim \mathcal{N}(0, \sigma^2 I)$, and let $Z \perp\!\!\!\perp G$. Let $b_1, \ldots, b_v$ be a finite a collection of basis functions and set $b(t) = (b_1(t), \ldots, b_v(t))^\top$. Then set

$$Q^*(t) = Z \cdot b(t) + G(t)$$

where $\sigma^2, \rho_G > 0$ are parameters to be estimated. The basis $b$ will be problem-specific and could be a polynomial basis, Fourier basis, or any number of other bases depending on context. The case $v = 1$ with a constant intercept is closely related to *ordinary kriging* and the case $v > 1$ is closely related to *universal kriging* [66, p. 8]. The apparent redundancy in the parameterisation due to the product $\sigma^2 \rho_G$ will be explained later. Using the notation $t = (t_1, \ldots, t_p)$, we consider a tensor product covariance model $k_G(t, t') = \prod_{i=1}^p k_{G,i}(t_i, t_i')$, $k_{G,i}(t_i, t_i') = \phi_i (\|t_i - t_i'\|/\ell_{t,i})$, for some radial basis functions $\phi_i$, scaled to satisfy $\phi_i(0) = 1$, and length-scale parameters $\ell_{t,i} > 0$ to be estimated.

**Prior for $E$:**   The process $E(h, t)$ is a model for the numerical error $q(h, t) - q^*(t)$, $t > 0$, which may be highly structured. A flexible prior model is therefore required. Moreover the error will, by definition, depend on the order of the numerical method; for successful extrapolation we must therefore encode this order into the model for $E$. It was earlier argued that a stationary GP is inappropriate, since the error is assumed to be $\mathcal{O}(h^\alpha)$. However, we observe that $h^{-\alpha}(q(h, t) - q^*(t))$ is $\mathcal{O}(1)$, suggesting that this quantity can be modelled using a stationary GP. We therefore take $E \sim \mathcal{GP}(0, \sigma^2 \rho_E k_E)$, where $\rho_E > 0$ is a parameter to be estimated, and

$$k_E((h, t), (h', t')) = (hh')^\alpha \psi (|h - h'|/\ell_h) \cdot k_G(t, t') \tag{4}$$

for a radial basis function $\psi$, scaled to satisfy $\psi(0) = 1$, and a length-scale parameter $\ell_h > 0$ to be estimated. Note how (4) separates the $h$ and $t$ dependence of $E$ in the prior, and adopts the same covariance model $k_G$ that was used to model the $t$ dependence of $G$. This can be motivated by the alternative perspective that follows from observing that $Q$ is a GP with covariance function

$$k_Q((h, t), (h', t')) = \sigma^2 \left\{ b(t) \cdot b(t') + \rho_G k_G(t, t') \left( 1 + \rho_E \frac{k_E((h, t), (h', t'))}{k_G(t, t')} \right) \right\} \tag{5}$$

where $k_E/k_G$ is a kernel only depending on $h$. Written this way, the model is seen to perform universal kriging over $T$ with a covariance adjusted by a multiplicative error arising from non-zero values of $h$.

**Higher-order convergence:**   The GP specification just described is not arbitrary; it ensures that the higher-order convergence property of RDAL is realised in BBPN. Consider again the setting in Proposition 1. Suppose that there exist $L, \varepsilon_0 > 0$ and $\beta \in (0, 1]$ such that $|1 - \psi(\varepsilon)| \leq L \varepsilon^\beta$ for all $\varepsilon \in [0, \varepsilon_0)$. Then the posterior mean $\mathbb{E}[Q(0)|D_h]$ satisfies $|q^* - \mathbb{E}[Q(0)|D_h]| = \mathcal{O}(h^{\alpha+\beta})$ as $h \to 0$. Thus if $\psi$ is Lipschitz (*i.e.* $\beta = 1$), BBPN achieves the same higher-order convergence, $\alpha + 1$, as RDAL. In this context we recall that any Matérn covariance function of smoothness at least $1/2$ is Lipschitz. The proof is provided in Appendix B.2.

**Model parameters:**  The free parameters of our prior model, $\sigma^2$, $\rho_G$, $\rho_E$, $\ell_h$, and the $\ell_{t,i}$ for $i = 1, \ldots, p$, are collectively denoted $\theta$. For all experiments in this article, $\theta$ was set using the maximum likelihood[2] estimator $\theta_{\mathrm{ML}}$. Our choice of parameterisation ensures that the maximum likelihood estimate for the overall scale $\sigma^2$, denoted $\sigma^2_{\mathrm{ML}}$, has a closed form expression in terms of the remaining parameters. This is analytically derived in Appendix B.3 and can be plugged straight into the likelihood. Gradients with respect to the remaining $3 + p$ parameters are derived in Appendix B.3, and gradient-based optimisation of the log-likelihood was implemented for the remaining parameters.

**Remark 3.** *GP interpolation, as with classical RDAL, is not parameterisation invariant. Thus some care is required to employ a parameterisation of $h$ that is amenable to the construction of a GP interpolant. The effect of differing parameterisations is explored in Appendix C.2.*

**Remark 4.** *The classical definition of RDAL presupposes that, in order to employ the method, the order $\alpha$ must be known a priori [67]. However if $\alpha$ is not known, the probabilistic perspective affords us the opportunity to learn $\alpha$ as an additional parameter in the statistical model—a procedure with no classical analogue. The feasibility of learning $\alpha$ is explored in Section 4.2.*

**Code:**  Software for BBPN, including code to reproduce the experiments in Section 4, can be downloaded from `github.com/oteym/bbpn`.

## 4  Experimental Assessment

This section reports a rigorous experimental assessment of BBPN. Firstly, Section 4.1 demonstrates that BBPN is competitive with existing PN methods in the context of *ordinary differential equation*s (ODEs). This result is somewhat surprising, given the black box nature of BBPN compared to the bespoke nature of existing PN methods for ODEs. Secondly, in Section 4.2 we demonstrate the versatility of BBPN by applying it to the nonlinear problem of eigenvalue computation, for which no PN methods currently exist. Finally, in Section 4.3 we use BBPN to provide uncertainty quantification for state-of-the-art numerical methods that aim to approximate the solution of nonlinear PDEs.

**Default Settings:**  We use Matérn(1/2) kernels for $\phi_i$ and $\psi$, *i.e.* $\phi_i(t_i, t_i') = \exp(-\|t_i - t_i'\|/\ell_{t,i})$, and similarly *mutatis mutandis* for $\psi$. These kernels impose a minimal continuity assumption on $q$ without additional levels of smoothness being assumed. Sensitivity of results to the choice of kernel is investigated in Appendix C.2.

**Performance Metrics:**  PN is distinguished from traditional numerical analysis by its aim to provide probabilistic uncertainty quantification, but nevertheless approximation accuracy remains important. To perform an assessment on these terms, we considered two orthogonal metrics. Firstly, we compute the *error* of the point estimate (mean), denoted $W \coloneqq \|\mathbb{E}[Q(0,\cdot)|D] - q^*(\cdot)\|$, where the norm is taken over $t \in T'$ where $T'$ is either $T$ itself or a set of representative elements from $T$. Secondly, and most importantly from the point of view of PN, we consider the *surprise* $S \coloneqq \|\mathbb{C}[Q(0,\cdot)|D]^{-1/2}(\mathbb{E}[Q(0,\cdot)|D]) - q^*(\cdot))\|$, where $\mathbb{C}[Q(0,\cdot)|D]$ denotes the posterior covariance matrix. If the true quantity of interest $q^*$ was genuinely a sample from $Q(0,\cdot)|D$, then $S^2$ would follow a $\chi^2$ distribution with $|T'|$ degrees of freedom. This observation enables the *calibration* of a PN method to be assessed [68]. Both metrics naturally require an accurate approximation to $q^*$ to act as the ground truth, which is available using brute force computation in Sections 4.1 and 4.2 but not in Section 4.3. The role of Section 4.3 is limited to demonstrating BBPN on a problem class that is challenging even for state-of-the-art methods.

### 4.1  Ordinary Differential Equations

The numerical solution of ODEs has received considerable attention in PN, with several sophisticated methods available to serve as benchmarks. Here we consider numerical solution of the following Lotka–Volterra IVP, a popular test case in the PN literature:

$$\frac{\mathrm{d}\mathbf{y}}{\mathrm{d}t} = f(t, \mathbf{y}) = \left[ \begin{array}{c} 0.5y_1 - 0.05y_1 y_2 \\ -0.5y_2 + 0.05y_1 y_2 \end{array} \right], \quad \mathbf{y}(0) = \left[ \begin{array}{c} 20 \\ 20 \end{array} \right]$$

---

[2]Alternative approaches, such as cross-validation, could also be used; see Chapter 5 of [57]. Our choice of maximum likelihood was motivated by the absence of any degrees of freedom (such as the number of folds of cross-validation), which permits a more objective empirical assessment.

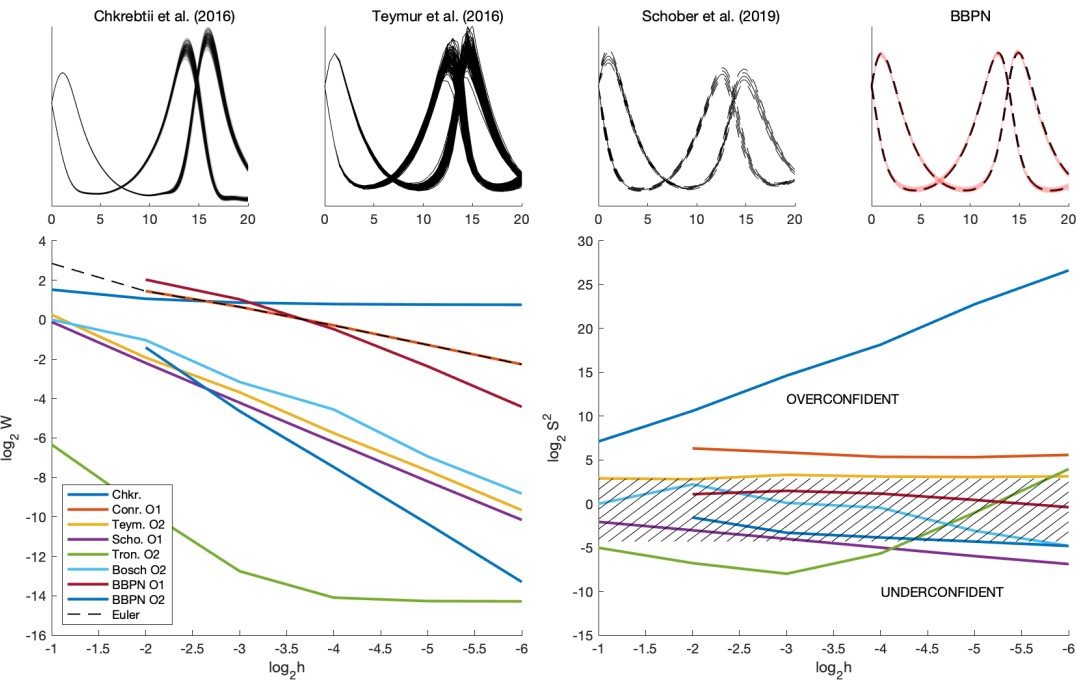

Figure 2: Ordinary differential equations. Top: Output from three existing PN algorithms [39]–[41] and BBPN, applied to the Lotka–Volterra IVP. Bottom left: The error $\log_2 W$ at the final time point $t_{\text{end}} = 20$, as a function of the time step size $h$. Bottom right: The surprise $\log_2 S$ at $t_{\text{end}} = 20$, with the central 95% probability band of a $\chi_2^2$ random variable shaded. Methods shown with (where applicable) their order: Chkr. [39]; Conr. O1 [38]; Teym. O2 [41]; Scho. O1 [40]; Tron. O2 [47]; Bosch O2 [55]; BBPN O1 & O2; and (traditional) Euler.

The aim in what follows is to approximate the quantity of interest $q^* = \mathbf{y}(t_{\text{end}})$ for $t_{\text{end}} = 20$. The top row of Figure 2 displays output from three distinct PN methods due to [39]–[41], as well as BBPN. (For these plots the coarse step-size $h = 0.5$ was used, so the probabilistic output can be easily visualised.) In each case, these methods treat a sequence of evaluations of the gradient field $f$ as data which are used to constrain a random variable model for the unknown solution of the ODE. Their fundamentally differing character makes direct comparisons challenging, particularly if we are to account for computational cost. However, each algorithm has a recognisable discretisation parameter $h$, so it remains instructive to study their $h \to 0$ limit. (In most cases $h$ represents a time step size, but the method of [55] is step-size adaptive; in this case $h$ is an error tolerance that is user-specified.) The methods of [38], [39], and [41] require parallel simulations to produce empirical credible sets, and thus have a significant computational cost. The methods of [40], [47] and [55] are based on Gaussian filtering and are less computationally demanding, though in disregarding nonlinearities the description of uncertainty they provide is not as rich. Interestingly, the output from [39] becomes overconfident as $h \to 0$, with $S^2$ being incompatible with a $\chi_2^2$ random variable, while the output from [40] becomes somewhat pessimistic in the same limit. Aside from these two outputs, the other PN methods considered appear to be reasonably calibrated.

To illustrate BBPN, our data consist of the final states produced by either an Euler (order 1) or an Adams–Bashforth (order 2) algorithm, which were performed at different resolutions $\{h_i = 2^{-i}, i = 1 \dots, 6\}$. The dataset[3] is augmented cumulatively, so that for $i = i'$, all data generated by runs $1, \dots, i'$ are used. The finest resolution in each case, $h_i$, is simply denoted $h$. For this experiment we use a prior with constant intercept, *i.e.* $v = 1$ and $b_1(t) = 1$. The BBPN output, shown in the bottom row of Figure 2, is observed to be calibrated, and the (order 2) output provides the most accurate approximation among all calibrated PN methods considered. Note in particular how BBPN accelerates the convergence of the Euler method from first order to second order, akin to RDAL.

---

[3]In this experiment the two components of $q^*$ were treated as *a priori* independent, but this is not a specific requirement of BBPN and dependence between outputs can in principle also be encoded into the GP model.

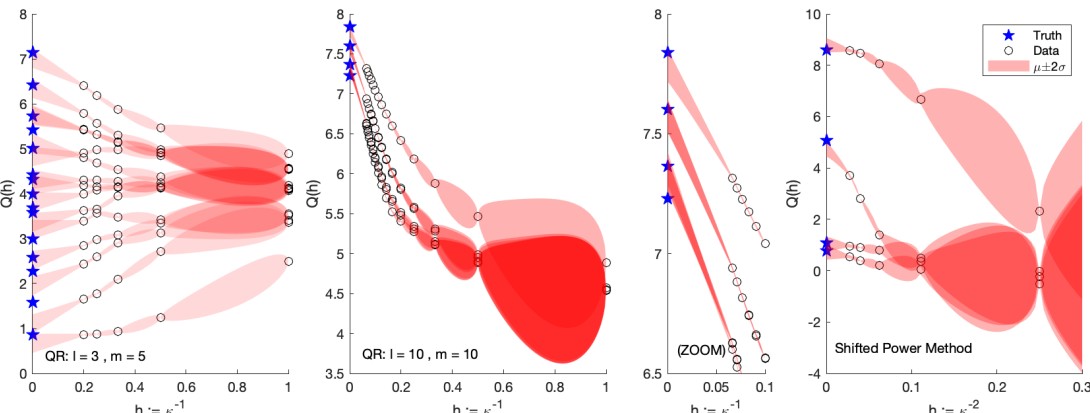

Figure 3: Eigenvalue problems. Left and centre: QR algorithm. Right: Shifted power method. Details of each simulation are given in the main text. All plots show red shaded $\pm 2\sigma$ credible intervals, numerical data as black circles, and true eigenvalues as blue stars.

In terms of computational cost, BBPN requires running a traditional numerical method at different resolutions, as well as the fitting of a GP. In this experiment, the computational cost of BBPN was intermediate between the filtering approach of [40] and the sampling approaches of [39] and [41]. Further details, including the sources of all these codes, are given in Appendix C.3.

## 4.2 Eigenvalue Problems

The calculation of eigenvalues is an important numerical task that has not yet received attention in PN. In this section we apply BBPN to (1) the QR algorithm for matrix eigenvalue discovery, and (2) a cutting-edge adaptive algorithm for the *tensor* eigenproblem, called the shifted power method [69]. In these examples, the order $\alpha$ is *unknown* and we append it to $\theta$ as an additional parameter to be estimated using maximum likelihood.

**QR Algorithm:** Take $T = \{1, \ldots, n\}$. Let $w \in \mathbb{N}$ and define $h := w^{-1}$. Given a matrix $A \in \mathbb{R}^{n \times n}$, its Schur form $A_\infty$ is approximated by matrices $A_w := R_{w-1}Q_{w-1}$, where $R_{w-1}$ and $Q_{w-1}$ arise as a result of performing a QR decomposition on $A_{w-1} = Q_{w-1}R_{w-1}$, and where $A_0 := A$. Then $q(h, \cdot)$ is the vector $\text{diag}(A_{h^{-1}})$, and $q^* = \text{diag}(A_\infty)$ is the vector of eigenvalues of $A$. As a test problem, whose eigenvalues are available in closed form (see Appendix C.4), we consider the following family of sparse matrices that arise as the discrete Laplace operator in the solution of the Poisson equation by a finite difference method with a five-point stencil. Let the $l \times l$ matrix $B$ and the $ml \times ml$ matrix $A$ be defined by

$$B = \begin{pmatrix} 4 & -1 & & \\ -1 & 4 & -1 & \\ & \ddots & \ddots & -1 \\ & & -1 & 4 \end{pmatrix}, \qquad A = \begin{pmatrix} B & -I & & \\ -I & B & -I & \\ & \ddots & \ddots & -I \\ & & -I & B \end{pmatrix}.$$

BBPN output for this problem is displayed in Figure 3. In the left-hand pane, we take $l = 5, m = 2$ and perform 5 QR iterations, displaying all 10 eigenvalues. In the centre pane, we take $l = 10, m = 10$ and perform 15 iterations. For clarity, this pane only displays the largest few eigenvalues of this $100 \times 100$ matrix, and we also show a zoomed-in crop to better demonstrate the extrapolation quality. Both examples show the convergence of $Q(h, \cdot)$ to $q^*(\cdot)$ as $w \to \infty$. Recall that $\alpha$ is *inferred* in these simulations —the maximum likelihood values were, respectively, 1.0186 and 1.0167.

The extrapolation performed by our GP model is seen visually to be effective and almost all true eigenvalues are contained within the $\pm 2\sigma$ credible intervals plotted. For comparison, in Appendix C.2 we contrast the result of using a *stationary* GP model (*i.e.* $\alpha = 0$). The extrapolating properties of that GP are immediately seen to be unsatisfactory, and we support this observation by examining the calibration of the two approaches, in a similar manner to in Section 4.1. This analysis strongly supports our proposed GP specification in Section 3.3.

**Shifted Power Method:** This iterative algorithm, due to [69] and implemented in [70], finds (random) eigenpairs of higher-order tensor systems, and we include it to demonstrate BBPN on a challenging problem in linear algebra. For $\boldsymbol{\mathcal{A}}$ a symmetric $m$th-order $n$-dimensional real tensor, and $\mathbf{x}$ an $n$-dimensional vector, define

$$\left(\boldsymbol{\mathcal{A}}\mathbf{x}^{m-1}\right)_{i_1} \coloneqq \sum_{i_2=1}^{n} \cdots \sum_{i_m=1}^{n} a_{i_1 i_2 \ldots i_m} x_{i_2} \ldots x_{i_m}, \qquad i_1 = 1, \ldots, n,$$

and say that $\lambda \in \mathbb{R}$ is an eigenvalue of $\boldsymbol{\mathcal{A}}$ if there exists $\mathbf{x} \in \mathbb{R}^n$ such that $\boldsymbol{\mathcal{A}}\mathbf{x}^{m-1} = \lambda \mathbf{x}$ and $\mathbf{x}^\top \mathbf{x} = 1$. Here we take $n = m = 6$ and produce a random symmetric tensor using the `create_problem` function of [70]. Two parameterisations of $q$ were considered; $h \coloneqq w^{-1}$ and $h \coloneqq w^{-2}$, where $w$ denotes the number of iterations performed, with results based on the latter parameterisation presented in the right-hand pane of Figure 3. (The maximum likelihood value for $\alpha$ in this example was 1.3318.) It can be seen that, after 5 iterations, BBPN is more accurate than each of the individual approximations on which it was trained. The choice of parameterisation affects the performance of BBPN, an issue we explore further in Appendix C.2.

In this example there is no additional computational cost to BBPN in the data collection stage, since the dataset is generated during a single run of an iterative numerical method. Therefore the only overhead is due to fitting the GP; though for this example this cost is itself negligible. All details, including a systematic assessment of error $W$ and surprise $S$ as $h$ is varied for both the QR algorithm and the shifted power method, are given in Appendix C.4.

**Remark 5.** *In this section we have implicitly modelled eigenvalues as* a priori *independent, for simplicity of exposition. Heuristics from random matrix theory suggest that, when treated probabilistically, eigenvalues may be better modelled with some non-trivial dependence structure. We note that this additional structure can be easily incorporated into the prior GP model for BBPN.*

## 4.3 Partial Differential Equations

To demonstrate the potential of BBPN on a challenging problem for which state-of-the-art numerical methods are required, we consider numerical solution of the *Kuramoto–Sivashinsky equation* [71, 72]

$$\partial_t u + \partial_x^4 u + \partial_x^2 u + u \partial_x u = 0. \tag{6}$$

This equation is used to study a variety of reaction-diffusion systems, producing complex spatio-temporal dynamics which exhibit temporal chaos—the characteristics of which depend strongly on the amplitude of the initial data and the domain length. We consider a flat initial condition $u(x, 0) = \exp(-0.01x^2)$ with periodic boundary condition on the domain $0 \leq x \leq 1$, and numerically solve to $t = 200$. Our quantity of interest is therefore $u(x, 200)$ for the domain $x \in [0, 1]$.

To obtain numerical solutions to (6) we transfer the problem into Fourier space and apply the popular fourth-order time-differencing ETD RK4 numerical scheme; see Appendix C.5 and [73]. ETD RK4 was designed to furnish fourth-order numerical solutions to time-dependent PDEs which combine low-order nonlinear terms with higher-order linear terms, as in (6). Computing accurate approximations to the solution of chaotic, stiff PDEs is a challenging problem for existing PN methods because computationally demanding high-order approximations across both spatial and temporal domains are required. Here, we assess BBPN applied to three sequences of five runs of ETD RK4, with minimum temporal step size $h = \delta t$ and, for simplicity, a fixed spatial step size $\delta x = 0.001$ throughout.[4] A reference solution was generated by taking $h = 0.0005$, but this cannot of course be guaranteed to be an accurate approximation to the true solution of (6).

Results shown in Figure 4 are encouraging; not only can accurate approximations be produced, but the associated uncertainty regions appear to be reasonably well calibrated, insofar as the magnitude of the uncertainty is consistent with the magnitude of the discrepancy between the posterior mean and the reference solution. Full details of these simulations are contained in Appendix C.5.

---

[4]For the $h = 0.002$ simulation in Figure 4, we have $h_i \in \{0.002, 0.005, 0.01, 0.02, 0.05\}$, for the $h = 0.005$ simulation we have $h_i \in \{0.005, 0.01, 0.02, 0.05, 0.1\}$, and for the $h = 0.01$ simulation we have $h_i \in \{0.01, 0.02, 0.05, 0.1, 0.2\}$

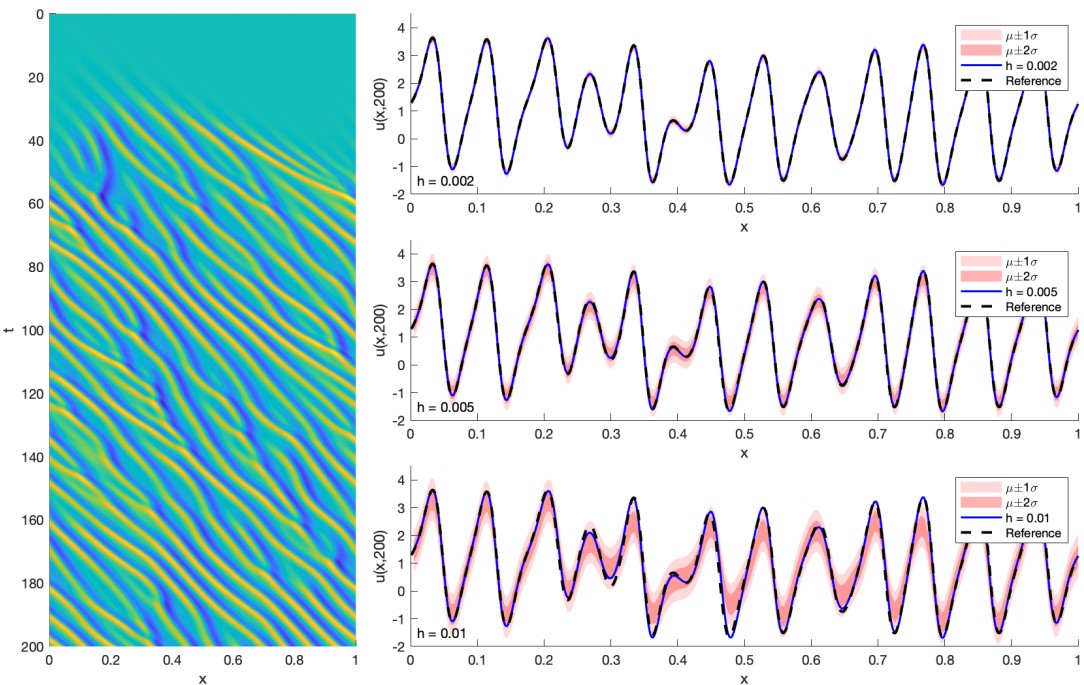

Figure 4: Partial differential equations. Left: Solution to the Kuramoto–Sivashinsky equation. Right: Approximation of the solution at the final time point ($t = 200$) using BBPN, based on minimum time step sizes $h \in \{0.002, 0.005, 0.01\}$. Posterior mean (blue) and credible regions (shaded) are displayed. A reference solution (dashed black) is obtained by taking $h = 0.0005$.

## 5  Discussion

This paper presented *black box probabilistic numerics*, a simple yet powerful framework that bridges the gap between existing PN methods and the numerical state-of-the-art. Positive results were presented on the important problems of numerically approximating ODEs, eigenvalues, and PDEs. Our main technical contribution is a probabilistic generalisation of Richardson's deferred approach to the limit, which may be of independent interest.

The main drawbacks, compared to existing PN, are a possibly increased computational cost and the additional requirement to model the error of a traditional numerical method. Compared to existing PN, in which detailed modelling of the inner workings of numerical algorithms are exploited, only the order of the numerical method is used in BBPN (and we can even dispense with that, as in Section 4.2), which may reduce its expressiveness in some settings. However, despite the black box approach, BBPN was no less accurate than existing PN in our experiments, and in fact the higher-order convergence property may enable BBPN to out-perform existing PN.

Some avenues for further research (that we did not consider in the present article) include the use of more flexible and/or computationally cheaper alternatives to GPs, the adoption of principles from experimental design to sequentially select resolutions $h_i$ given an overall computational budget, and the simultaneous use of different traditional (or even probabilistic) numerical methods within BBPN.

## Acknowledgments and Disclosure of Funding

This work was supported by the Lloyd's Register Foundation programme on data-centric engineering at the Alan Turing Institute, UK. The authors wish to thank Jon Cockayne and Ilse Ipsen for feedback on earlier versions of the manuscript, and Nicholas Krämer for assistance with the `probnum` package.

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
