## Supplementary Material

These appendices contain supplementary material for the paper *Black Box Probabilistic Numerics*.

## A    Proof of Proposition 1

The equation of the straight line through two points $(x_1, y_1)$ and $(x_2, y_2)$ is given by

$$\frac{y - y_1}{x - x_1} = \frac{y_2 - y_1}{x_2 - x_1}.$$

Substituting the points $(h^\alpha, q(h))$ and $((\gamma h)^\alpha, q(\gamma h))$, and taking $x = 0$, we have

$$y = q(h) - \frac{q(\gamma h) - q(h)}{\gamma^\alpha - 1}.$$

By the assumption that $q$ is of order $\alpha$, we have the expansions $q(h) = q^* + Ch^\alpha + \mathcal{O}(h^{\alpha+1})$ and $q(\gamma h) = q^* + C(\gamma h)^\alpha + \mathcal{O}(h^{\alpha+1})$, and then by substitution and straightforward cancellation we find

$$y = q^* + \mathcal{O}(h^{\alpha+1}).$$

Therefore the $y$-intercept of the line is an approximation of $q^*$ of order $\alpha + 1$.

## B    Gaussian Processes for BBPN

This appendix contains full details of how analytic conditioning formulae are obtained and how maximum likelihood estimates are calculated.

### B.1    Conditioning Formulae

It will be convenient to introduce *lexicographic ordering*, where the indices

$$\{(i,j) : j = 1, \ldots, m_i, \ i = 1, \ldots, n\} \tag{7}$$

are ordered first by $i$ and then, for indices with the same $i$, by $j$. Let $h_{(l)}$ and $t_{(l)}$ denote, respectively, the values of $h_i$ and $t_{i,j}$ corresponding to the $l$'th ordered pair $(i,j)$ in (7). Let $\mathbf{q}$ represent a column vector of length $m := \sum_{i=1}^n m_i$, with entries $\mathbf{q}_l := q(h_{(l)}, t_{(l)})$ in lexicographic order.

From (5), the prior model for $Q$ described in Section 3.3 has covariance function

$$k_Q((h,t),(h',t')) = \sigma^2[b(t) \cdot b(t') + \rho_G k_G(t,t') + \rho_E k_E((h,t),(h',t'))], \tag{8}$$

where the additivity follows from the assumptions $Q^* \perp\!\!\!\perp E$ and $Z \perp\!\!\!\perp G$. Let $K_Q$ be an $m \times m$ matrix and $\mathbf{k}_Q(h,t)$ be an $m \times 1$ column vector with entries of the form

$$(K_Q)_{l,l'} := k_Q((h_{(l)}, t_{(l)}),(h_{(l')}, t_{(l')})), \quad (\mathbf{k}_Q(h,t))_l := k_Q((h_{(l)}, t_{(l)}),(h,t)). \tag{9}$$

Then standard Gaussian conditioning formulae (eg. Equation 2.19 in [57]) demonstrate that the conditional process $Q|D$ has mean and covariance functions

$$\mu_{Q|D}(h,t) = \mathbf{k}_Q(h,t)^\top K_Q^{-1} \mathbf{q} \tag{10}$$

$$k_{Q|D}((h,t),(h',t')) = k_Q((h,t),(h',t')) - \mathbf{k}_Q(h,t)^\top K_Q^{-1} \mathbf{k}_Q(h',t') \tag{11}$$

The mean and covariance functions of the marginal process $Q(0, \cdot)|D$ are extracted by setting $h$ equal to 0 in Equations (10) and (11).

### B.2    Proof of Higher-Order Convergence Result in Section 3.3

For a scalar quantity of interest, the full covariance function in (5) is

$$k_Q(h, h') = a_1 + a_2(hh')^\alpha \psi\left(\frac{|h - h'|}{\ell_h}\right)$$

for certain positive constants $a_1$ and $a_2$. For $\gamma \in [0, 1]$, denote

$$\psi_h = \psi\left(\frac{(1-\gamma)h}{\ell_h}\right).$$

Then the conditional mean at $h = 0$, given the data $D_h = \{(h, q(h)), (\gamma h, q(\gamma h))\}$, is

$$\mathbb{E}[Q(0)|D_h] = \begin{pmatrix} q(h) \\ q(\gamma h) \end{pmatrix}^\top \begin{pmatrix} a_1 + a_2 h^{2\alpha} & a_1 + a_2 \gamma^\alpha \psi_h h^{2\alpha} \\ a_1 + a_2 \gamma^\alpha \psi_h h^{2\alpha} & a_1 + a_2 \gamma^{2\alpha} h^{2\alpha} \end{pmatrix}^{-1} \begin{pmatrix} a_1 \\ a_1 \end{pmatrix}$$

$$= \frac{q(h)\gamma^\alpha(\gamma^\alpha - \psi_h) + q(\gamma h)(1 - \gamma^\alpha \psi_h)}{a_1 a_2(1 - 2\gamma^\alpha \psi_h + \gamma^{2\alpha})h^{2\alpha} + a_2^2 \gamma^{2\alpha}(1 - \psi_h^2)h^{4\alpha}} a_1 a_2 h^{2\alpha}$$

$$= \frac{q(h)\gamma^\alpha(\gamma^\alpha - \psi_h) + q(\gamma h)(1 - \gamma^\alpha \psi_h)}{a_1(1 - 2\gamma^\alpha \psi_h + \gamma^{2\alpha}) + a_2 \gamma^{2\alpha}(1 - \psi_h^2)h^{2\alpha}} a_1.$$

Inserting $q(h) = q^* + Ch^\alpha + \mathcal{O}(h^{\alpha+1})$ and $q(\gamma h) = q^* + C\gamma^\alpha h^\alpha + \mathcal{O}(h^{\alpha+1})$ in the above equation yields

$$\left| q^* - \mathbb{E}[Q(0)|D_h] \right| = q^* \left| 1 - \frac{\gamma^\alpha(\gamma^\alpha - \psi_h) + 1 - \gamma^\alpha \psi_h}{a_1(1 - 2\gamma^\alpha \psi_h + \gamma^{2\alpha}) + a_2 \gamma^{2\alpha}(1 - \psi_h^2)h^{2\alpha}} a_1 \right|$$

$$+ \left| \frac{\gamma^\alpha(\gamma^\alpha - \psi_h) + \gamma^\alpha(1 - \gamma^\alpha \psi_h)}{a_1(1 - 2\gamma^\alpha \psi_h + \gamma^{2\alpha}) + a_2 \gamma^{2\alpha}(1 - \psi_h^2)h^{2\alpha}} \right| |C| a_1 h^\alpha$$

$$+ \left| \frac{\gamma^\alpha(\gamma^\alpha - \psi_h) + 1 - \gamma^\alpha \psi_h}{a_1(1 - 2\gamma^\alpha \psi_h + \gamma^{2\alpha}) + a_2 \gamma^{2\alpha}(1 - \psi_h^2)h^{2\alpha}} \right| a_1 \mathcal{O}(h^{\alpha+1})$$

$$\leq q^* \left| \frac{a_2 \gamma^{2\alpha}(1 - \psi_h^2)}{a_1(1 - 2\gamma^\alpha \psi_H + \gamma^{2\alpha}) + a_2 \gamma^{2\alpha}(1 - \psi_h^2)h^{2\alpha}} \right| h^{2\alpha}$$

$$+ \left| \frac{\gamma^\alpha(1 + \gamma^\alpha)}{1 - 2\gamma^\alpha \psi_h + \gamma^{2\alpha}} \right| |C| |1 - \psi_h| h^\alpha$$

$$+ \mathcal{O}(h^{\alpha+1}).$$

It follows from the Hölder assumption $|1 - \psi(\varepsilon)| \leq L\varepsilon^\beta$ that $|1 - \psi_h| = \mathcal{O}(h^\beta)$. Therefore the second term, which dominates the right-hand side, is of order $\mathcal{O}(h^{\alpha+\beta})$. This concludes the proof.

### B.3   Maximum Likelihood Estimation

The parameters $\theta$ of the covariance function $k_Q$ are estimated from data using maximum likelihood. Recall that (with $\alpha$ known) $\theta$ consists of the parameters $\sigma$, $\rho_G$, $\rho_E$, $\ell_h$, and the $\ell_{t,i}$ for $i = 1, \ldots, p$. This parameterisation is deliberately chosen to enable the maximum likelihood estimator $\sigma_{\mathrm{ML}}$ to be computed as an explicit function of the remaining components of $\theta$. It is convenient to express

$$k_Q((h, t), (h', t')) = \sigma^2 \overline{k}_Q((h, t), (h', t'))$$

where $\overline{k}_Q((h, t), (h', t'))$ is (8) with $\sigma = 1$. Analogously define $\overline{K}_Q$ as in (9) but with $\sigma = 1$. The log-likelihood of observing the dataset $D$ in (2) under the model for $Q$ defined in (3) can then be expressed as

$$\mathcal{L}(\theta) = -\frac{m}{2}\log(2\pi) - m\log\sigma - \frac{1}{2}\log|\overline{K}_Q| - \frac{1}{2\sigma^2}\mathbf{q}^\top \overline{K}_Q^{-1}\mathbf{q}, \tag{12}$$

where we note that $\overline{K}_Q$ does not depend on $\sigma$ but can depend on all the other components of $\theta$. In the case of the overall amplitude parameter $\sigma$, it is possible to obtain an analytic expression for the value $\sigma_{\mathrm{ML}}$ by differentiating and setting $\partial\mathcal{L}/\partial\sigma = 0$ [74]. This gives

$$\sigma_{\mathrm{ML}}^2 = \frac{\mathbf{q}^\top \overline{K}_Q^{-1}\mathbf{q}}{m} \tag{13}$$

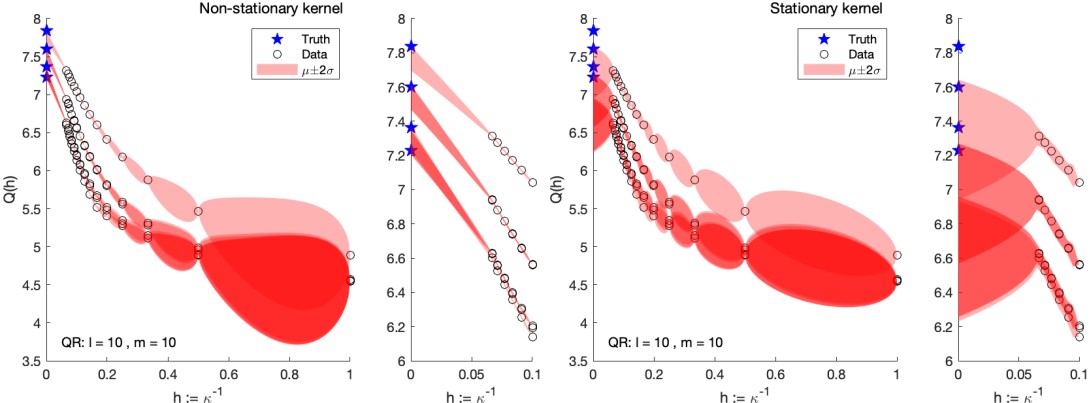

Figure 5: Comparison of stationary (4), on left, and non-stationary (16), on right, covariance functions for the QR algorithm example detailed in Section 4.2.

Plugging $\sigma = \sigma_{\mathrm{ML}}$ into (12) gives

$$\mathcal{L}(\theta|\sigma = \sigma_{\mathrm{ML}}) = -\frac{m}{2}\log(\mathbf{q}^\top \overline{K}_Q^{-1}\mathbf{q}) - \frac{1}{2}\log|\overline{K}_Q| + C \tag{14}$$

where $C$ is a constant in $\theta$. From here, we employ numerical optimisation to maximise (14) over the remaining $3 + p$ degrees of freedom in $\theta$.

It is important to ensure that numerical optimisation is successful, otherwise conclusions provided by BBPN could be an artefact of failure of the numerical optimisation method. To this end, we undertake robust gradient-based optimisation on (14), using MATLAB's packaged `fmincon` routine. This requires calculation of the gradients of (14) and explicit formulae will now be provided.

By differentiating (14) we have

$$\partial_\theta \mathcal{L}(\theta|\sigma = \sigma_{\mathrm{ML}}) = \frac{m}{2}\frac{\mathbf{q}^\top \overline{K}_Q^{-1}(\partial_\theta \overline{K}_Q)\overline{K}_Q^{-1}\mathbf{q}}{\mathbf{q}^\top \overline{K}_Q^{-1}\mathbf{q}} - \frac{1}{2}\mathrm{tr}\big(\overline{K}_Q^{-1}(\partial_\theta \overline{K}_Q)\big) \tag{15}$$

Define the matrices

$$(B)_{l,l'} := b(t_{(l)})\cdot b(t_{(l')}) \,, \quad (K_G)_{l,l'} := k_G(t_{(l)}, t_{(l')}) \,, \quad (K_E)_{l,l'} := k_E((h_{(l)}, t_{(l)}), (h_{(l')}, t_{(l')})).$$

Then $\overline{K}_Q = B + \rho_G K_G + \rho_E K_E$, and it follows that

$$\begin{aligned}
\partial_{\rho_G} K_Q &= K_G \,, & \partial_{\ell_h} K_Q &= \rho_E \partial_{\ell_h} K_E \,, \\
\partial_{\rho_E} K_Q &= K_E \,, & \partial_{\ell_{t,i}} K_Q &= \rho_G \partial_{\ell_{t,i}} K_G + \rho_E \partial_{\ell_t} K_E
\end{aligned}$$

The low-level terms such as $\partial_{\ell_h} K_E$ can readily be computed by hand and will depend on the radial basis functions $\phi_i$ and $\psi$ adopted in $K_G$ and $K_E$. Note that if $\alpha > 0$ is treated as unknown and appended to the parameter vector $\theta$, as in Section 4.2, a similar calculation can be performed to obtain the gradient with respect to $\alpha$ of (14).

The convergence of this gradient-based optimisation approach to a minimum of $\mathcal{L}(\theta)$ is verified empirically in Appendix C.3.2.

## C    Details of Empirical Assessment

This appendix contains full details for all experiments described in the main text.

### C.1    Riemann Sum Illustration in Figure 1

Figure 1 considers the function $f(x) = \sin^2(4\pi x) + \exp(x) - \frac{5}{2}x^4 + \frac{1}{2}\cos(16\pi x) + \frac{1}{4}\cos(20\pi x)$. The quantity of interest $q^*$ is the integral $\int_0^1 f(x)\,\mathrm{d}x$, which has the exact value $(\mathrm{e} - 1) \approx 1.71828$.

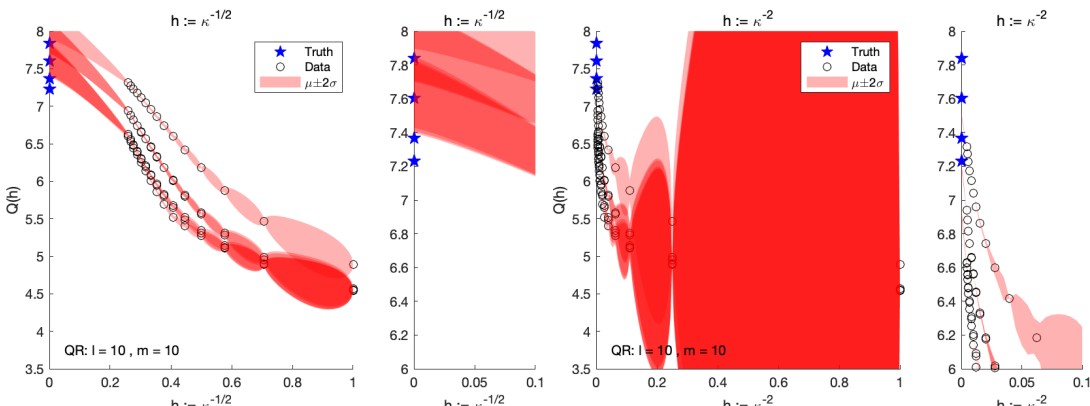

Figure 6: Comparison of different parameterisations for $h$ relative to the number of iterations $\kappa$ of the QR algorithm; $h := \kappa^{-1/2}$ (left); $h := \kappa^{-2}$ (right)

BBPN was applied to the method of Riemann sums. The convergence of this method is first order, and we set $\alpha = 1$ accordingly. We choose a range of step-sizes $h$ between $0.01$ an $0.08$, with the Riemann sum approximations plotted in the left pane of Figure 1. Hyperparameters of the GP were set using maximum likelihood approach, as described in Appendix B.3.

## C.2 Sensitivity to Prior Specification

In this section we consider the effect of varying several of the choices made during the specification of our prior model. The suitability of our non-stationary GP model is considered in Appendix C.2.2. The effect of the choice of parametrisation for $h$ is considered in Appendix C.2.2. The choice of the kernel functions $\phi_i$ and $\psi$ is discussed in Appendix C.2.3. Finally, the nature and number of the finite-dimensional basis terms $b_i$ is discussed in Appendix C.2.4. In each case we explore the impact of these aspects of the prior specification by reproducing figures from the main text under different settings within the GP model.

### C.2.1 Stationary / Non-Stationary Error Model

Since the error $E(h, t)$ is assumed to vanish in the limit $h \to 0$, and since its scale is assumed to depend on the order $\alpha$ of the underlying numerical method, we specified a non-stationary GP in (4). For the QR algorithm example in Section 4.2, we now contrast this with the same analysis performed with the stationary GP whose covariance function is

$$\tilde{k}_E((h,t),(h',t')) = \psi\left(|h - h'|/\ell_h\right) \cdot k_G(t, t') \tag{16}$$

*i.e.* setting $\alpha = 0$ in (4).

From Figure 5 (right), we see that the extrapolation is extremely poor when a stationary GP is used. Moreover, the use of a stationary GP leads in this case to over-confident predictions, with the true eigenvalues belonging outside of the $\pm 2\sigma$ credible intervals. This provides strong support for the use of the non-stationary GP that we propose in the main text.

### C.2.2 Parameterisation of $h$

The choice of parameterisation of $h$ is also crucial to the operation of BBPN. While it is sometimes the case that an 'obvious' parameterisation exists (such as the step-size in a time-stepping method, where the order $\alpha$ specifically refers to this quantity; or the overall tolerance level of a numerical method) this is, unfortunately, not always true. If some heuristic reasoning for determining this parameterisation is not available, we recommend some prior experimentation and comparison with calibration metrics such as surprise, introduced in Section 4.

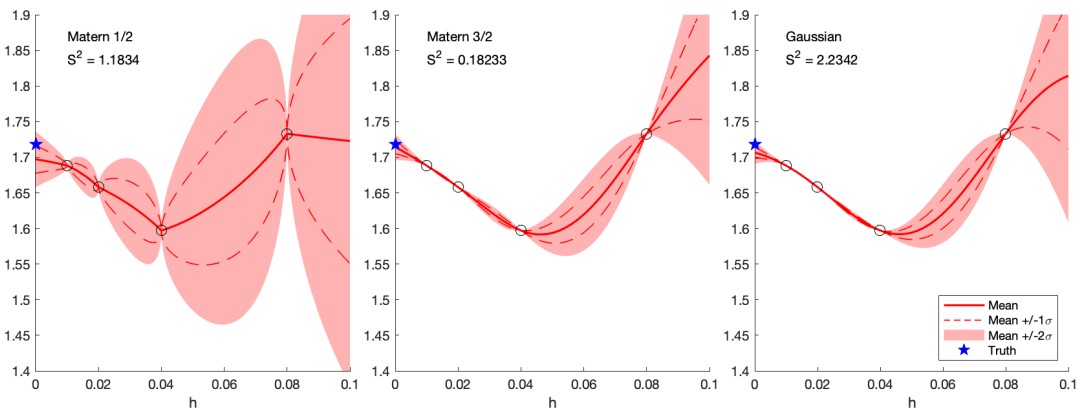

Figure 7: Comparison of different kernel types for the radial basis functions $\phi_i$ and $\psi$. Matérn 1/2 (left); Mateérn 3/2 (centre); and Gaussian (right).

For the QR algorithm example in Section 4.2, Figure 6 shows the effect of replacing the parameterisation $h := \kappa^{-1}$ (as in Figure 3) with $h := \kappa^{-1/2}$ and $h := \kappa^{-2}$. Although BBPN continues to work, to an extent, with these alternative parametrisations, its predictive performance is somewhat diminished.

### C.2.3 Choice of Radial Basis Functions $\phi_i$ and $\psi$

For all simulations in this article we specified Matérn 1/2 kernels for $\phi_i$ and $\psi$. The motivation for this, stated in the preliminary notes in Section 4, is to impose the minimal continuity assumption on $q$ but not to assume additional levels of smoothness where this cannot be justified *a priori*.

Figure 7 shows the effect of specifying instead Matérn 3/2 or Gaussian kernels for $\phi_i$ and $\psi$ in the Riemann sum test problem in Figure 1, contrasting with the Matérn 1/2 kernel used there. In all cases, the same process of gradient-based optimisation was employed to automate the setting of the kernel hyperparameters. The additional smoothness of the mean interpolant is clearly visible in the higher Matérn and Gaussian cases, but note also the difference in scale of the $\pm 2\sigma$ region. In particular, the use of smoother kernels is associated with higher confidence in the predictive output, with the Gaussian kernel producing the largest value of the surprise $S^2$ (though this was still within the central 95% region for a $\chi^2$ distribution, so we do not reject the hypothesis that the BBPN output is calibrated). On balance we err on the side of caution and recommend the Matérn 1/2 kernel for applications of BBPN.

### C.2.4 Choice of Basis Functions $b_i$

In this section we demonstrate the purpose of including basis functions $b_i$ in the model for $G(t)$. To do so, we plot the output of the BPPN procedure for the PDE example in Figure 4, since this example has non-trivial '$t$' domain (though the variable called $t$ in the model definition in Section 3 is in fact called $x$ here). The effect of including a constant basis function (*i.e.* $v = 1$ and $b_1(t) = 1$) is to allow the model a non-zero mean in $t$. For this example, the dynamics are mostly above the $0$ level and even a simple global mean would be more likely between $1$ and $2$. Omitting the basis function (*i.e.* $v = 0$), as shown in the bottom pane of Figure 8, inflates the covariance to compensate for this misfit, and in this case results in an underconfident model.

In this example, it is unlikely that the additional inclusion of higher-order polynomial basis functions would be of use. Indeed our experiments showed this. However the oscilliating nature of the dynamics across the range of $t$ suggests a Fourier basis may be an appropriate mean model. Ideas along these lines are partially explored in [75], and a fuller investigation in the context of BBPN will be the subject of future work.

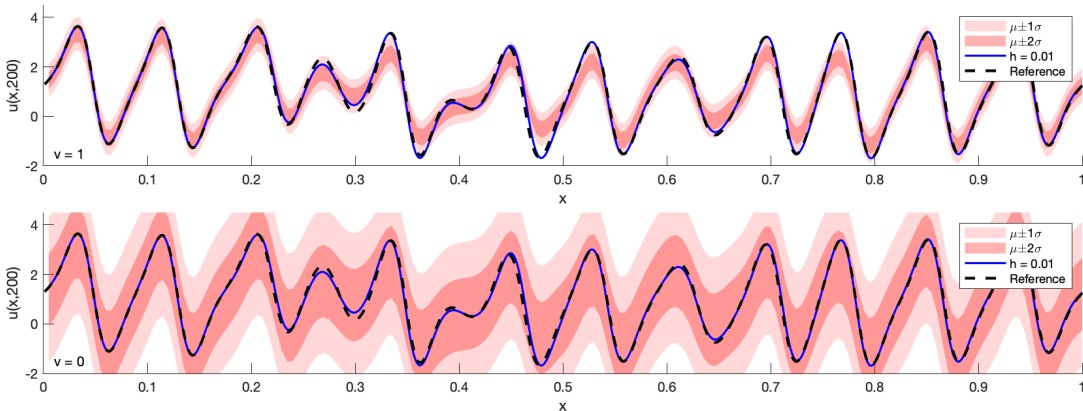

Figure 8: Comparison of the inclusion and exclusion of the first polynomial basis function (top: $v = 1$, bottom: $v = 0$) for the model in Section 4.3.

## C.3 Ordinary Differential Equations

Here we provide full details for the ODE experiment in the main text. In Appendix C.3.1 we explain how all the probabilistic ODE solvers that we considered in the main text were implemented. Then, in Appendix C.3.2, we present evidence that the gradient-based optimisation approach we employed to estimate the GP hyperparameters in BBPN has successfully converged.

### C.3.1 Details of Implementation

In this section we describe in detail the sources and licences of the codes, as well as the settings used, to perform the comparison experiments in Section 4.1. These codes are from different sources, span several years in release date, and are coded in different languages. They also accept inputs and give outputs in mutually inconsistent forms. This makes a 'cloned-repository' solution from which results could be reproduced automatically impractical. In the interests of maximum possible transparency we manually collect and present code sources and parameter values here in the hope that interested readers will not find it difficult to reproduce our results locally if required. Recall that our simulations consist of varying input $h$.

The one-step-ahead sampling model of Chkrebtii *et al.* [39] (labelled 'Chkr.' in Figure 2) was run using `MATLAB` code from `https://git.io/J33lL` with `nsolves` $= 100$, `N` $= \lceil 20/h \rceil$, `nevalpoints` $= 1001$ and the `lambda` and `alpha` hyperparameters left at their default values (which depend on $\mathbb{N}$, and therefore $h$). This software has no explicitly-stated licence.

The perturbed integrator approach of Conrad et al. [38] and Teymur et al. [41] (labelled 'Conr. O1' and 'Teym. O2' was run using `MATLAB` code provided to us by the authors of the latter paper and not, as far as we are aware, publicly released.

The Gaussian filtering approach of Schober et al. [40], Tronarp et al. [47] and Bosch et al. [55] (labelled 'Scho. O1', 'Tron. O2' and 'Bosch O2') was run by installing the `Python` package `probnum` and using the function `probsolve_ivp`. 'Scho. O1' uses non-adaptive step-sizes and takes `algo_order` $= 1$, and `method` $=$ `EK0`; 'Tron. O2' uses non-adaptive step-sizes and takes `algo_order` $= 2$, and `method` $=$ `EK1`; while 'Bosch O2' uses *adaptive* step-sizes and takes `algo_order` $= 2$, and `method` $=$ `EK1`. In the latter case, $h$ is taken as the relative tolerance `rtol` instead of the step-size. This software is Copyright of the ProbNum Development Team and is released under an MIT licence.

The reference solution used in calculating errors was calculated using `MATLAB`'s in-build `ode45` function with tolerances set using `odeset('RelTol',3e-14,'AbsTol',1e-20)`

It is difficult to fairly compare the wall-clock times of these codes, particularly since they are written in different languages and are therefore run in different environments. For the example simulation in Figure 2, none of the examples took more than a few seconds on a 2018 MacBook Pro, and some were virtually instant. All publicly-available codes were downloaded or cloned on 22 April 2021.

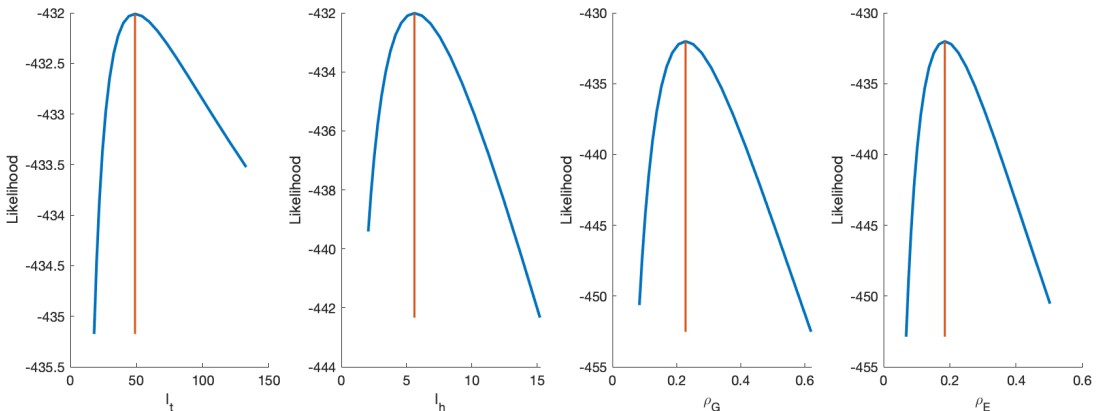

Figure 9: Likelihood variation in the neighbourhoods of the maximum likelihood values found by MATLAB's `ode45` optimiser. In each case, the remaining parameters were fixed at their maximum likelihood value. The values determined by the optimiser are shown with a vertical orange line.

### C.3.2 Parameter Identifiability

In order to assess the robustness of our gradient-based optimisation procedure for maximum likelihood estimation, we consider again the Lotka–Volterra model. Here we will vary each parameter $\ell_t$, $\ell_h$, $\rho_G$ and $\rho_E$ in turn, holding all other parameters fixed at the values produced by the gradient-based optimisation method. The resulting plots are given in Figure 9.

In this application, at least, we can be reasonably confident that the optimisation procedure has located a global maximiser in 4D (though, strictly speaking, we cannot confirm this from the univariate plots in Figure 9). In general, and as is common in GP modelling, model fit should always be assessed, in order to be confident in the data-driven nature of the GP output, something particularly salient in numerical applications where calibration is of paramount importance.

### C.4 Eigenvalue Problems

In this section we provide certain further details for the eigenvalue problem presented in Section 4.2. We first note that the matrix $A$ defined there can be shown to have exact eigenvalues $4 - 2\cos(p\pi/(l+1)) - 2\cos(q\pi/(m+1)); p = 1, \ldots, l; q = 1, \ldots, m$. The knowledge of the true values is required to facilitate the following analysis. For this section we take $l = 3$ and $m = 5$, as in the left-hand panes of Figure 3.

In a similar manner to Figure 2, we plot in the left-hand pane of Figure 10 the (log-) error $W$ for several methods —the classic QR algorithm in green, then the traditional extrapolation methods of Richardson and Bulirsch–Stoer (using the data obtained in the run of the QR algorithm) in red and yellow respectively. From the definition of $W$, this 'combined absolute error' is formed by considering the norm of the error vector of *all* eigenvalues. (The centre pane gives the (log-) maximum relative error $w$, *i.e.* $\max_i[(\hat{\lambda}_i - \lambda_i)/\lambda_i]$, where $\hat{\lambda}$ is the vector of true eigenvalues, and is provided since this is a more familiar presentation of error in eigenvalue problems in numerical analysis.)

It is seen that polynomial and even rational function interpolation are not robust in this setting, and give errors significantly larger than simply the most accurate single QR-produced estimate. BBPN does not suffer the same issue, possibly because the nonparametric interpolant has favourable stability properties, and it is somewhat competitive with the traditional QR algorithm, at the cost of additional computation but with the additional richness of output that a PN method provides.

The right-hand pane shows the (log-squared-) surprise of *individual* eigenvalues of the $15 \times 15$ matrix, plotted over the 95% central probability region of a $\chi_1^2$ random variable. This shows that the predictions provided for the majority of the 15 eigenvalues are well-calibrated, but that a small number of predictions are overconfident. This is a promising early result for a problem with no previous PN method in existence, as well as one in which $\alpha$ has to be inferred due to the absence of a canonical parameterisation for $h$; see Appendix C.2.2.

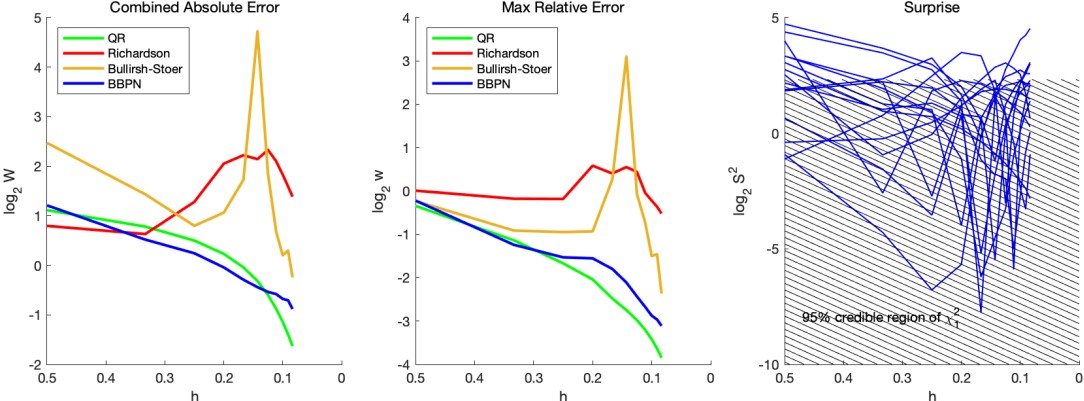

Figure 10: Eigenvalue Problems: Left: the combined absolute error $W$ for the classic QR algorithm (green), the traditional extrapolation methods of Richardson (red) and Bulirsh–Stoer (yellow) using the data obtained in the run of the QR algorithm, and BBPN (blue). Centre: the maximum relative error $w$ for the same methods. Right: the surprise $S$ of *individual* eigenvalues of the $15 \times 15$ matrix, plotted over the 95% central probability region of a $\chi_1^2$ random variable.

## C.5   Kuramoto–Sivashinsky Equation (KSE)

In this section we provide further detail for the PDE problem presented in Section 4.3.

Numerical solutions to the *Kuramoto–Sivashinsky equation* (KSE) were computed on the spatial grid $x \in \{0, 0.001, 0.002, \dots, 1\}$ and over time segments $t_{i,j} = jh_i$ for $j \in \{0, 1, \dots, m_j\}$, where $m_j = \lfloor 200/h_i \rceil$ with $\lfloor \bullet \rceil$ denoting the nearest integer function and $h_j$ the time-step parameter. After transformation into Fourier space, solutions were computed using a fourth-order Runge–Kutta numerical integrator ETDRK4 [73].

### C.5.1   Fourier Transform to Employ the ETDRK4 Numerical Integration Scheme

We discretise the spatial domain using a Fourier spectral transformation. That is, we set

$$u(x,t) \approx \sum_{k \in \Omega_k} \tilde{u}_k(t) \exp^{ikx/L},$$

in (6), where $\Omega_k$ denotes the set of wave-numbers. Doing so returns the Fourier transformed KSE,

$$\frac{d}{dt}\tilde{u}_k(t) + \left(\frac{k^4}{L^4} - \frac{k^2}{L^2}\right)\tilde{u}_k(t) + \frac{ik}{2L}\tilde{v}_k(t) = 0, \qquad t > 0, \tag{17}$$

where

$$\tilde{v}_k(t) = \frac{1}{2\pi L}\int_{-\pi L}^{\pi L} u^2(x,t)\exp^{-ikx/L}\,dx \approx \frac{1}{N}\sum_{l=0}^{N-1} u^2(x_l, t)\exp^{-ikx_l/L}$$

with $N = 1/\delta x$ and $\delta x$ denoting the spatial step-size, and on assuming that both the solution and spatial derivative are periodic in $x$, *i.e.*,

$$u(x,t) = u(x + 2\pi L, t) \quad \text{and} \quad \frac{\partial}{\partial x}u(x,t) = \frac{\partial}{\partial x}u(x + 2\pi L, t), \qquad t \geq 0,$$

for some user defined length scale $L$ (which we take to be $L = 1/2\pi$ in our simulation). See [73] for a complete description of the fourth-order ETDRK4 scheme, as well as example MATLAB code used to compute solutions to (17).