# OpenReview forum: "Black Box Probabilistic Numerics"
_NeurIPS.cc/2021/Conference — NeurIPS 2021 Poster_

### Official Review · Reviewer_reNR · 2021-07-13

**Rating:** 7
**Confidence:** 2

**Summary:**

This paper presents a probabilistic version of Richardson's "deferred approach to the limit" (RDAL) method.  RDAL is an established numerical analysis method (developed in 1927) with many applications. In RDAL several low-accuracy approximations are extrapolated to produce a more accurate approximation (that is costly to come with directly).
In the framework proposed in this draft, namely "Black box probabilistic numerics" (BBPN), the aforementioned extrapolation/curve fitting is carried out via Gaussian Processes and therefore comes with all the benefits for Bayesian approaches such as quantification of the  estimation uncertainty. This approach is simple and it can potentially be combined with Bayesian methods other than GPs as well.
In the experiment section the following applications are presented: nonlinear ODE/PDE i.e. ordinary/partial differential equations and eigenvalue problem.


**Limitations And Societal Impact:**

Yes,they mention that the limitations of their proposed algorithm are: increased computational costs as well as additional assumptions required for modelling uncertainty.
Since this work is theoretical it does not have a direct societal impact.



**Main Review:**

The main idea of this paper boils down to probabilistic curve fitting via Gaussian Processes. This idea is simple, up to my knowledge original and potentially significant.  Nevertheless, I think the contribution and impact  of this paper could be enriched if the authors would conduct more thorough experiments that quantitatively compare BBPN against state-of-the-art. In the current draft, only in the first experiment such a quantitatively comparison is provided.



**Time Spent Reviewing:**

4

---

> ### Author Response · Authors · 2021-08-10
> **Response to reNR**
>
> We thank the reviewer for a positive review. We chose to make our comparisons with PN ODE methods because these are among the most well studied PN methods. Note that since the submission we have additionally made some revisions to the output of Figure 2 that better reflects the ‘state of the art’ in the case of the set of comparisons with the ‘probnum’ suite of algorithms. This presents a fairer comparison with the most advanced recent methods in that package, including adaptive methods, and these will be included in the revision along with an updated discussion to reflect the latest ‘state-of-the-art’.

---

### Official Review · Reviewer_fzBQ · 2021-07-14

**Rating:** 7
**Confidence:** 3

**Summary:**

This paper proposes a new "black box" approach to probabilistic numerics (PN)
that leverages existing (non-probabilistic) numerical methods to estimate,
probabilistically, the true, intractable quantity of interest (e.g. the solution
of an ODE or of an eigenvalue problem). The method is demonstrated in three
experiments and is (i) compared to other PN approaches for solving ODEs, (ii)
applied to eigenvalue problems (for which no PN method has yet been developed),
and (iii) used for uncertainty quantification on a challenging nonlinear PDE
problem. Due to its generality, the proposed method expands probabilistic
numerical approaches to a larger range of problems.


**Limitations And Societal Impact:**

Limitations are discussed in Section 5 (Discussion) and Section 1
(Introduction):
- BBPN has possibly increased computational cost (due to the multiple calls to
  the numerical method that BBPN requires);
- BBPN requires a statistical model of the numerical error;
- BBPN only makes use of the order of the method - in comparison, other non-BB
  PN methods could potentially benefit from considering additional information
  about the numerical method.

In addition, the important dependence on the parameterisation of h is discussed
in more detail in the supplementary C.2.2. An additional experiment where h is
not equal to a step size, e.g. for ODE solvers with adaptive step-size
selection, might provide additional relevant experimental evidence.
In a similar light, comparisons to other PN methods on problems without such a
clear notion of step-size, such as quadrature or linear systems, could provide
further supporting evidence.



**Main Review:**

__Originality__
The presented BBPN method presents, to the best of my knowledge, a completely
novel approach to the field of PN.
The relation to the well-established Richardson extrapolation is made very
clear, and relevant PN approaches for the considered problems seem adequately
cited.


__Quality__
The submission is technically sound and the paper provides all necessary
derivations.
The experimental results cover an appropriate range of experiments, though a
wider range of comparisons to a wider range of PN methods that go beyond ODE
solvers, e.g. for quadrature or solving linear systems, would provide
additional, useful evidence to best assess the empirical qualities of the
proposed approach.
Similarly, the flexibility could be further demonstrated by more experiments
that do not have such a clear notion of "step size" as in the ODE and PDE
example.
But again, while such additions would be very welcome I want to emphasize that
the current evaluation is already very useful and covers a good range of
experiments.


__Clarity__
The presentation is very clear. The paper provides relevant background on the
Richardson extrapolation and on PN and presents the method in an understandable
and concise manner. The appendix provides many helpful details.


__Significance__
I believe that the paper provides a significant contribution to the field of
probabilistic numerics. Due to the universal applicability of the presented
method, it potentially provides a new baseline for any future PN method that is
developed for novel problem classes (e.g. for eigenvalue problems as mentioned
in the paper).


**Time Spent Reviewing:**

6

---

> ### Author Response · Authors · 2021-08-10
> **Response to fzBQ**
>
> We thank the review for a supportive review. Any notion of ‘tolerance’ (eg. in the error of a quadrature estimate, or the normed error in the solution of a linear system) can be used in place of the step size as our parameter h, and the method still operates as described. We do make brief reference to this on Line 118 but will clarify this in the revision.

---

### Official Review · Reviewer_7xVs · 2021-07-16

**Rating:** 7
**Confidence:** 4

**Summary:**

This paper introduces a black-box approach to construct a probabilistic numerical method by treating a sequence of increasingly accurate approximations generated by a classic numerical method as observations. This probabilistic analogue of Richardson's deferred approach to the limit allows higher orders of convergence to be achieved and extends the PN approach to previously unexplored tasks.

**Limitations And Societal Impact:**

## Limitations
To my mind, the presented approach has the following limitations, which lead to some questions, which as a reader I felt were not entirely answered:
- As the authors state themselves BBPN requires an underlying numerical method to be run to convergence for a sequence of parameters h. This makes BBPN (and RDAL) roughly a constant factor more expensive than simply running a classical method. When is this tradeoff worth it for the gain in accuracy? Also, likely there are settings (e.g. linear dependence on quantity of interest on data), where using PN methods which condition directly on the observations of the problem are much cheaper. I realize that wall-clock times are difficult to compare across implementations. However, running the same numerical method multiple times plus the added cost of GP inference in comparison to the cost of a single run to the highest accuracy is valid to my mind and would give some insight here.
- Performing regression on a dataset with few observations (outputs of numerical methods) is a setting where Gaussian processes are well-established. However, in high-dimensional settings and when performing extrapolation, their performance may be quite poor. To my understanding this corresponds to the case where one cannot run the baseline numerical method to high-precision or the quantity of interest is high-dimensional (e.g. eigenvalue problems for realistically sized matrices n >> 100). Does this limit BBPN to problems where the QoI is low-dimensional?
- In the same vein, inferring the order of the numerical method (\alpha), while an attractive feature, may be computationally quite costly relative the benefit it provides. Also from the paper i was not able to judge how well the convergence order was inferred, since for the presented eigenvalue estimation problem the true convergence order is not known.

## Societal Impact
No direct negative impact due to the methodological nature of this paper.

**Main Review:**


The idea of extending Richardson's deferred approach to the limit (RDAL) by conditioning a stochastic process on the output of a sequence of numerical methods seems very natural once one accepts the central premise of PN that epistemic uncertainty is to be quantified. I therefore find the approach presented in this paper to be quite attractive. Overall the paper is well-executed both in terms of its coherence, mathematical presentation, as well as experimental evaluation. There are some inherent limitations that come with the choice of method in my opinion and some important questions that are left somewhat unexplored. However, overall the positives by far outweigh the limitations, which is why I would recommend acceptance. See below for details.

## Novelty
From a classical perspective the added functionality of a certain class of interpolating functions given by the posterior mean of the GP that satisfy Proposition 1, may appear somewhat limited, since they directly extend the existing idea behind RDAL. However, from a PN perspective the added functionality of uncertainty quantification for the quantity of interest is useful. In particular probabilistic numerical methods for linear algebra have been almost exclusively focussed on the solution of linear systems, while this approach naturally extends to the estimation of eigenvalues (Section 4.2). Also inferring the order of a numerical method via hyperparameter optimization is an interesting and novel functionality to the best of my knowledge, albeit in practice possibly of limited importance (see limitations below).

## Theoretical Contribution
While the underlying result of obtaining a higher-order of convergence is clearly not new, the authors are very upfront about this. The kernel construction which preserves this property however is elegant and presents a significant contribution.

## Experiments
The experimental evaluation of the presented BBPN approach is well-presented and also includes calibration. Often in practice PN methods suffer from miscalibration due to the problem of non-linear dependence of the quantity of interest on the data. The experimental demonstrations of how BBPN applies to different problems in numerics is commendable. However, clearly these experiments are illustrative in nature and may mask some of the limitations of the approach, in particular for high-dimensional quantities of interest (i.e. eigenvalue estimation for matrices of n >> 100) or expensive numerical methods, where computational budget does not allow running the method more than a few times.

## Clarity and Reproducibility
The paper is generally well-written with a clear line of exposition, supported by explanatory figures. The paper comes with code reproducing some of the figures and has sufficiently detailed experimental descriptions in the appendix.

### Typos
- l30: computational resources

**Time Spent Reviewing:**

5

---

> ### Author Response · Authors · 2021-08-10
> **Response to 7xVs**
>
> We thank the reviewer for a comprehensive review.
>
> We agree that there will clearly be cases (such as a linear quantity of interest, as identified by the reviewer) where PN methods designed for generality will be outperformed by specialised approaches. The main motivation behind this work is ‘black box’ settings in which such simplifications cannot be assumed. We envisage situations in which the quantity of interest is related to the data in a highly complex and non-linear fashion, but that nevertheless numerical algorithms are available that are able to give output when run at different fidelities.
>
> GPs certainly have limitations in high dimensions. This in fact opens up some interesting points. If for example some additional dependence structure is modelled *across* dimensions (something we do not do but which the multi-output model can trivially include) then this problem may not be so severe. For things like the eigenvalue problem it is not totally obvious what approach to take, but eg. heuristics from random matrix theory suggest that eigenvalues may be better modelled as non-independent if treated probabilistically. We are pursuing some of these issues in current work, and we will make clearer reference to them in the revision.
>
> Thank you again for your thoughtful response.

---

> > ### Comment · Reviewer_7xVs · 2021-08-11
> > **Update after Author Response**
> >
> > Thank you for the clarifying response. My recommendation to accept this paper remains the same.

---

### Official Review · Reviewer_TEPo · 2021-07-16

**Rating:** 7
**Confidence:** 3

**Summary:**

The authors present a probabilistic numeric algorithm (BBPN) which operates analogously to Richardson's deferred approach to the limit (RDAL) in traditional numerical analysis, and has an advantage over other PN approaches in that it is applicable in any setting where an existing numerical method is available. The authors provide a detailed description and analysis of BBPN, including theoretical guarantees on the order of convergence. They finish with an experimental assessment of BBPN, by comparison to both probabilistic and traditional numerical methods.

**Limitations And Societal Impact:**

I do not see any potential negative societal impact, and the authors adequately address the limitations of their work.

**Main Review:**

This paper is a valuable, well-presented and very well-written contribution to the field of probabilistic numerics. The paper is well structured, striking a good balance between background, motivation, theory and experiment. BBPN is well motivated and clearly described, and the experimental results are convincing. I especially appreciated the presence of the uncertainty quantification metric in the experimental comparison against other probabilistic numeric methods for ODEs. I would welcome more discussion of the uncertainty quantification properties of BBPN earlier in the paper, since uncertainty quantification is of critical importance for probabilistic numeric methods. I understand that there may be no good way to expand this beyond what is already present in the experimental section, given the tight nine-page constraint.

**Time Spent Reviewing:**

3.5

---

> ### Author Response · Authors · 2021-08-10
> **Response to TEPo**
>
> We thank the reviewer for a generous review. We will signpost the fact that we attempt a UQ analysis earlier on in the paper since we agree that this is of crucial importance in PN.

---

### Decision · Program_Chairs · 2021-09-27

**Decision:**

Accept (Poster)

**Comment:**

This paper suggests a "black box" approach to probabilistic numerics. The idea is to treat the output of existing (non-probabilistic) numerical methods as observations, after which the true/unknown quantities can be treated via probabilistic inference. Examples are given applying this idea to ODEs, to eigenvalue problems, and to uncertainty with nonlinear PDEs. All reviewers felt the paper was novel, technically sound, useful, and clearly written.